# Active and Assisted Living, a Practice for the Ageing Population and People with Cognitive Disabilities: An Architectural Perspective

**DOI:** 10.3390/ijerph20105886

**Published:** 2023-05-19

**Authors:** Santiago Quesada-García, Pablo Valero-Flores, María Lozano-Gómez

**Affiliations:** 1Department of Architectural Design, University of Seville, Av. Reina Mercedes 2, 41012 Seville, Spain; mlozanomez@gmail.com; 2Faculty of Medicine, Campus Teatinos, University of Malaga, Blvr. Louis Pasteur, 32, 29010 Malaga, Spain; pvalero@uma.es

**Keywords:** Active Assisted Living, ambient intelligence, Ambient Assisted Living, health humanities, healthy architecture, ageing, ambient adapted to Alzheimer’s outpatients, cognitive architectural design

## Abstract

The current digital revolution is causing a paradigm shift encompassing all environments in which human beings conduct their daily activities. Technology is starting to govern the world, gradually modifying not only individual and social behaviour, but also ways of living. The necessary adaptation to new information and communication technologies forces societies to rethink both public and private spaces, in which evolution is slower than rapid social transformation. As part of this change, the concept of Active Assisted Living (AAL) has developed. Assisted spaces can be designed to provide older adults, carers, or people who have cognitive disabilities, such as Alzheimer’s disease or other dementias, with a healthier, safer, and more comfortable life, while also affording them greater personal autonomy. AAL aims to improve people’s quality of life and allow them to remain in their own homes for as long as possible, not in residences. This study conducted a critical review about AAL from an architectural point of view. The research adopted a qualitative approach in which we collected the studies during the last twenty years, then used descriptive, narrative and critical analysis methods. Based on these, this paper aims to explain this new technological paradigm, its characteristics, its main development trends, and its implementation limitations. The results obtained show how the development of AAL will be in the next ten years, and how this concept, and its application, can influence architecture and provide the bases for further research into the design of buildings and cities.

## 1. Introduction

Society is undergoing a rapid transformation in the way people transmit and receive information. The population is immersed in a silent digital revolution based on communication, which is changing the manner of social interaction, participation, residential solutions and even city models [1]. The communication revolution is having important consequences. Today, it is not necessary to go out into the street to shop, work, or debate a political issue; this can all be done through the Internet [2]. Information and communication technologies (ICTs) have emerged as an application of scientific knowledge for practical uses and have acquired a special importance.

Spaces are surrounded by interfaces that reside in all kinds of objects and are capable of continuously and invisibly recognising and responding to a person’s requests or physical presence. The current trend in technology is to be ubiquitous, transparent, and intelligent. On the other hand, the gap between people’s biological lives and the age of work and social retirement is increasing. This circumstance provides an opportunity to rethink how to live and plan for the important period in human lives entailed in ageing, which can span 20 or 30 years. Population ageing and the technological revolution comprise some of the main issues of concern that are currently at the centre of the architectural debate. Quality of life may be improved through interactions between people and technology as the latter adapts to individuals’ behaviour. This involves the careful planning and design of spaces, which allow for the sustainable and ethical inclusion of these emerging technologies, in such a way that they may provide confidence and security, while working against social exclusion.

This paper addresses, from the architectural point of view, the complex interactions and implications that technology will have in the ambient and environments of older adults and people with physical or cognitive disabilities so that they have a healthy, safe, and active ageing. These are the starting topics and questions of the study, and they will be developed in the third section of this article.

Architecture may integrate emerging technologies into building construction and city planning [3]. The impact of digital technology on ways of living is increasingly present in the architectural debate, focused on the need to design environments that enhance the human condition, in the face of visual, geometric, or formal issues. In this context, the notion of Active Assisted Living (AAL) appears in domestic spaces for the purpose of introducing technology-based services and assistance, which enable people to have a healthier, safer, and more comfortable life in their home environment, for as long as possible [4]. AAL began with a European programme in 2008 due to the need for a response to this new technological paradigm, as will be discussed below; however, a review of the technological concept of AAL, from the architectural point of view, is pending. The theorisation and conceptualisation of AAL is needed not only in terms of technological assistance and services but also considering all of the inherent aspects of an inhabited space. In turn, the meaning of a new spatial design paradigm must be explored: what it consists of and how it may respond to the changing needs of a growing and constantly evolving ageing population. The originality of this paper is aimed at that gap of information, and its main objective is to fill it, connecting the AAL characterisation with the field of architecture.

To address this objective, the methodology of the work is mixed. On one hand, in support of the idea that Active Assisted Living is an architectural practice, an analysis was made of associated terms in reviews and summaries in the techno-logical literature. A search, appraisal, analysis, and synthesis framework was used to examine the main review terms. On the other hand, the methodology uses experiences and projects that will help to contextualise the problem and visualise the results obtained in several studies and research projects. Therefore, this work has a theoretical character, with some examples that show how AAL has been implemented.

The results obtained are a consequence of the engagement with sources as well as re-cent scholarship, and they display AAL as a new paradigm of design that should not be limited to providing technology-based services, but one that offers architectural elements that can be proactively incorporated into the design of buildings and cities.

This work has been structured into six sections. This introduction is followed by a presentation of the background and context, and the question is addressed in a broad context to highlight the purpose of the study. This is followed by an explanation of the method used to carry out the critical review. The next section shows the results and explains how this new paradigm of AAL arose and the main forces behind it. From the existing literature, the AAL concept is described, providing an explanation of its guiding principles, as well as its first experiences and implementation. The fifth section contains the discussion and a critical appraisal, analysing whether approaches from these theories have worked. The last section indicates the main conclusions, signifying some visions of future habitats for an increasingly ageing population. The issues addressed in this work provide added value for scholarship, and especially in the architectural field, by allowing AAL to be understood as a new concept that can be incorporated into the design of buildings and cities.

## 2. Background and Context

As pointed out in the introduction, today’s society is ageing with a concomitant in-crease in cognitive disabilities related to mental health. In recent years, new concepts have emerged together with the digital revolution, in which the same society is involved. There are terms such as the fourth digital decade, Smart House, Smart Buildings, and smart cities. These issues make up the context in which the AAL programme is being developed, as we will see below.

### 2.1. Living and Ageing in the Fourth Digital Decade

Jonathan Swift, in his well-known Gulliver’s Travels, said that every man desires to live long, but no man wishes to be old. However, getting older is still the only way we have found to live long, as stated by Sainte-Beuve [5]. Longevity can be understood as the period of time by which a living being manages to survive longer than other individuals of its same species. Pascal Bruckner maintains that the possibility of extending life is one of the most extraordinary collective achievements of humanity [6]. It is the most palpable reflection of the advances achieved in the field of health. Longevity in human beings is a mirror to the success of the fight against some infectious diseases; mortality resulting from childhood illnesses, and childbirth complications; or those related to more advanced age groups. Hence, health has become an issue of social concern through ageing.

Ageing is a biological process by which living beings become old, involving a series of structural and functional changes that appear over time and are not the result of diseases or accidents [5]. Old age becomes a very valuable resource. An older adult needs to continue to be eager in their desire to enjoy the world and to undertake new goals. Bruckner argues that old age should be as active and healthy as possible. The current social structures are, however, based on the life expectancies that were common after World War II, when 50 was the threshold for what was considered the beginning of old age. Currently, men and women of 50 are in a similar condition to those aged 30.

According to the report on the ageing of the world’s population by the Population Division of the Department of Economic and Social Affairs of the United Nations [7], in 2050, one in five people will be older than 60, and the number in this age bracket will reach 2.1 billion, exceeding the number of youths between the ages of 15 and 24. In addition, almost 80% of older adults in the whole world will pertain to developing countries, thus indicating that agedness is not exclusive to first world countries but rather is a general phenomenon [7].

The accelerated rhythm of ageing of the world’s population will impact on the demand for goods and services as well as social protection, health, housing, education, transportation, information, and communications. It will also influence the work and financial market, family structures and intergenerational relationships, and, undoubtedly, the development of multilevel governance. Such a general impact will create a need for policies and significant urban transformations as, by the middle of the 21st century, 70% of the world’s population will live in cities, and this population will be substantially aged, as indicated above.

The EU4Health Programme of the European Union (2021–2027) underlines that, to adequately face old age and achieve the best well-being in European societies, systems must incorporate policies that promote health and prevent illness in order to decrease the impact on the economy, assistance, and social development. By 2070, one in three Europeans will be older than 65 [8]. The ratio of the active to inactive population will reach one active person to four inactive persons; this proportion is currently one to two. With this assessment, in recent years, the European Commission has developed a line of action that combines the uncertainty of the cost of old age, due to disease, dependency and other factors, with the appreciation of the added value that older adults bring, with their knowledge, experience, and so on.

Thus, the growing paradigm of healthy ageing arises, based on developing and maintaining functional, physical, and cognitive capacities in a way that enables the well-being of people during their ageing process. This is achieved if they can perform, fully and autonomously, the instrumental activities of daily living [9] while fulfilling their needs and life aspirations. People’s functional, physical, and cognitive skills are influenced by both physiological and psychological changes and by aspects related to health, or the presence or absence of illness. These capacities are determined by two factors that interact with one another. The first is the intrinsic faculty of each person, that is, the combination of their physical and mental skills. The second is the physical environment and social ambience in which they live [10].

### 2.2. New Architectural and Urban Models: Smart House, Smart Building, Smart City

Physical spaces and environments, as well as the technology integrated within them, can be planned and designed to adapt to people’s aptitudes and capacities. One of the most important actions to promote healthy old age is, therefore, the planning of adapted and adaptable environments. It is necessary to build a habitat, both individual and collective, which promotes the maximum quality of life, over the course of the constant evolution of human needs, throughout childhood, youth, maturity and, especially, in the context of ageing [11].

This means thinking about architecture and the cities of the future, with a view to them meeting the individual and social demands generated by this growing demographic trend. It also means giving priority to essential aspects for older adults, such as experience and memory, accessibility and adaptation, functionality and diversity, integration and social interaction, as well as the meaning and use of habitable spaces and environments (Figure 1). These are intangible values that must be added to the growing technological influence, caused by the gradual and unstoppable digital transformation that has surrounded contemporary society over the last three decades and is still far from being complete [12].

This digital revolution, dizzying and silent, is leading to changing attitudes in the ways in which people and environments interact, changing living habits and the production of a definite change of mentality within current society [2]. Every day, people are more intensely and proactively seeking, or demanding, constant improvements to their quality of life by means of efficient environmental, health, and intelligent solutions that must be as sustainable, affordable, and integral as possible. The digital revolution incorporates innovative forms of communication that promote new methods of interaction, through mobile devices, cell phones, television, and the Internet and entail a radical mutation in the relationships between people, and between people and domestic objects. The common element in the majority of everyday devices is their technological essence. They are devices are capable of gathering and processing data, which, thanks to the possibility of communicating with each other, makes them intelligent.

This new interaction is characterised by three factors: a ubiquitous experience in high-definition; the connection of electronic and home appliance devices with a range of services; and interactive communication between computers, telephones, tablets, and visual and touch recognition devices [13]. Technology has become another component in the landscape of everyday life.

The catalyst for this digital revolution has been the Internet, which is a decentralised set of interconnected communication networks that enable and build a fluid information transmission network. The concept of the Internet of Things (IoT), a phrase coined by Kevin Ashton, first appeared in 1999 at the MIT Auto-ID Center [14]. The IoT is a network of physical objects with embedded technology, such as sensors or software, which are capable of communication and interaction with each other, or with the external environment. In a home, these objects are typically household appliances or technical items (thermo-stats, home security cameras and systems, lighting fixtures, etc.), which can be controlled through devices, such as smartphones or voice-activated speakers. Actually, with regard to the integration of IoT devices and systems within the home, there are already open and freely available platforms, such as Openhab, Homeassistant, etc. Recently, as initiated by Apple, the GAFA (Google, Apple, Facebook, Amazon) multinational telecommunication companies have also presented their own concept addressing related issues.

The connectivity of these new products and platforms allows them to have some capabilities that are external to the physical device itself. These include the collection of data that can be analysed to inform decision-making, enable operational efficiency, and continually improve product performance. When these objects are able to change their responses based on the data information and their own experience, they are called ‘smart objects’, or ‘smart connected things’ (SCoT). A smart object is not only a connected element but also a product with an effective interconnected management, which improves its interaction with other objects and with people. It can be created as a manufactured product, or by embedding electronic chips or sensors, in nonsmart physical objects. SCoTs are active products, integrated by software, sensors, and a connectivity that permits data to be exchanged between the product, maker, operator/user, and other systems.

In buildings, the concept of IoT technology is applied to products used within a ‘SmartHome’. This type of home is made up of intelligent components, which are combined and reproducible in standardised models, and smart objects, which are designed to assist the domestic environment in real time through the synergetic interaction of automation systems and technologies, materials, and innovative processes. To achieve this goal, an inhabited place must have a set of elements, or subsystems, interconnected with each other through mutual relationships with open, secure, and highly reliable software acting together according to general, pre-established rules.

The response provided to the individual and collective needs of people by these assisted environments, progressively transforms into demands for effective, sustainable, and affordable services, both for the open market and cities. An osmotic and permeable process emerges, between the offer of services and the implementation of technologies, in which the micro, the intimate home space, enriches the macro, the urban space, and vice-versa. That is, there is an interaction between the domestic scale of the Smart House, the intermediate scale of the Smart Building, and the construction of a city must become increasingly more intelligent each day [15].

With these new technologies, the Smart City appears as a new urban model, open in its definition. In it, the city is understood not as a physical, closed, and geographically defined element, but rather as a network of a series of overlapping layers, or grids that interact in order to improve their functioning and efficiency. The basic layer of a Smart City is the environment, on which other networks are superimposed, consisting of the following grids, layers, or systems:Information analysis and management;Water supply and purification;Waste and productive management;Energy generation and supply;Urban mobility.

Each one of these grids corresponds to an urban system, which, in turn, consists of a multitude of interwoven nodes (Figure 2). These systems, which are equipped with sensors, are in communication with each other through a network that allows information management and decision making. The purpose of this organisation of the network, multiscale or fractal, is to facilitate governance and to efficiently respond, in real time, to the constant flow of social demands. The Smart City is a network of networks that changes, mutates, and is in constant movement. Based on the persistent needs of society, it also adapts, incorporates new layers, modifies existing ones, builds new networks, and creates unsuspected landscapes [13]. In a few years’ time, the services necessary to satisfy the demands and needs of the increasingly ageing population will constitute a new layer of the Smart City that must be designed and planned.

## 3. Methods

This work adopts the method presented by review articles, in which an analytical framework is used, following a process of search, appraisal, synthesis, and analysis [16]. This paper is based on qualitative research and performs a critical review of the existing literature with the aim of presenting, analysing, and synthetising the material from diverse sources. It seeks to achieve a global vision and to identify the most significant items in the field of Architecture.

In this critical review, the process examines the information on the item, then goes on to provide a full and objective view of what had hitherto been published. An analysis of the information is then made, seeking a critical evaluation, as well as highlighting if any area of knowledge is in need of further research. Moreover, within this method, it is possible to embody existing theories and present conceptual innovation from other fields, such as engineering.

A series of databases have been used for this bibliography review, and their choice responds to the interest that the different areas of knowledge have raised on these subjects. These databases are Web of Science, Scopus, PubMed and Science Direct. Moreover, databases specified in the architectural field, such as Avery (from Columbia University of New York) and RIBA (from the Royal Institute of British Architects) have been employed.

Once the databases were selected, concepts and keywords were set out for the search step, firstly registering all the publications in which the concept/keyword were included; secondly, an appraisal of the abstract and title was made; and finally, the literature obtained was filtered to produce a list for analysis. The concepts and keywords were the following: Active Assisted Living; ‘Active Assisted Living’; Active Assisted Living Ageing Population; Active Assisted Living Cognitive Disabilities; Active Assisted Living Smart Cities; Ambient Assisted Living; and ‘Ambient Assisted Living’. In addition, other books and articles were selected for their special significance to the issue as they form part of the common bibliography of the aforementioned literature.

In addition to the above process, a more precise filter was carried out with regard to the areas of knowledge, in which engineering, architecture, and medicine, stand out. Moreover, some databases enable the identification of a subdivision of fields from technical discipline: architecture, telecommunication, computer science and engineering. The Medical discipline was broken down into: Healthcare Science Services, Psychology, Behavioural Sciences, Geriatrics and Gerontology, Public Environmental Occupational Health and Neurology and Neurosciences. Figure 3 shows a model of the method carried out when choosing the Web of Science database. This same process was accomplished with the other previously mentioned databases, with a list of selected concepts and keywords.

During the execution of the above method, some outcomes could be perceived as an advancement of the results section. As a consequence of the scoping searches, a selection of the main publications were examined, and it quickly became apparent that a huge amount of literature mentioned the main terms. Nevertheless, just a few publications present the concept or syntagm Active Assisted Living. Around 40% of the existing publications came from the Medical discipline, and more than 45% of those were from a technical standpoint. However, in technical disciplines focused on the field of architecture and related to the AAL concept, which is still in its infancy, there is no canonical work to date. The literature selected and evaluated in the architectural field was mainly composed of articles with open access and high impact, which can be found in the bibliography of this paper. This review article is a starting point and identifies a potential opportunity for contemporary research.

The outcomes obtained made it necessary to filter the publications according to their dates (Figure 4). This decision showed that the syntagm Active Assisted Living arose in 2008 when the European Union launched a programme based on this concept. As a result, this work aimed to examine the scientific literature indexed in databases from the last 15 years. This filter by date determined a rapid increase of interest on the issue in the last few years; therefore, the analysis was focused on publications made during this recent period. Moreover, 2014 is marked as an important year as it was when the concept was retermed, from Ambient and Assisted Living to Active Assisted Living, which will be clarified later in this article. In addition, the literatures from 2006 to 2008 were taken into account as they present the germ of the concept.

## 4. Results

Following the method described above, several AAL research projects and practical applications have been detected. Some of these recent experiences have generated impact publications, in which their main achievements have been disseminated. It is these works with impact that are analysed and presented as case studies in this article, and from which the main results are obtained and are presented below.

### 4.1. The Emergence of New Concepts: Ambient Intelligence (AmI) and Active Assisted Living (AAL)

The application of new technological paradigms and means of management, to the field of healthcare and older adults, stemmed from the concept of ambient intelligence (AmI), at the end of the 1990s. The notion of AmI was originally developed by Eli Zelkha and Brian Epstein, and their team at Palo Alto Ventures [17], and was presented at the Digital Living Room Conference, organised by Philips, and developed during the following decade (Figure 5).

AmI refers to the use of intelligent assistance systems, embodied and integrated into different habitable environments, which create an omnipresent technological layer capable of transparent interaction with the inhabitant, observing and interpreting their actions and intentions. The objects are capable of proactive interaction with people where they are needed and are sensitive to both the user and the context (situational, spatial and temporal). The communication between the system and the user takes places in such a way that the device can act invisibly with the person and display a proactive behaviour, by which it takes action in advance of a future situation rather than reacting. These assistance systems are designed to take control of a situation and to make early changes, rather than adjusting to a situation or waiting for something to happen.

The AmI approach implies that technology is designed by and for people, rather than expecting the users to adapt to the technology, as has tended to be the case until now [19]. This is the main difference between IoT and Ambient Intelligence. AmI faces its application with a holistic view focused on the person. The concept of holistic (from the Greek *holos*, meaning whole or integral) refers to the notion that the whole is not the sum of the parts, nor can it be explained by analysing how its separate components function; rather, it is imperative to do so integrally, contemplating the reality of its complexity. A holistic perspective implies overcoming the prejudice of the incapable individual and replacing it with the notion of the real person [20]. The solutions of AmI aim to fulfil seven of the foundational principles of Universal Design, Design for All in Europe [21]. This is a process of designing products (devices, environments, systems and processes) that can be used by people with the widest range of skills, in the widest range of situations (environments, conditions and circumstances).

Also, AmI is not limited to a straightforward, unique, and connected relationship between objects or between the product and the user but rather extends to all of the components of the multidimensional space (ambient), surrounding the inhabitant, the floor, walls, ceiling, paths, spaces, clothes, etc. [22]. The ambient begins to be ‘intelligent’ when it does not need to be calculated or programmed. The environment can learn the preferences of the inhabitant, constantly adapting the parameters of the system and, therefore, improving the comfort, well-being, and quality of life of the inhabitant [23]. Ambient Intelligence offers a new paradigm in which people can use technology through an environment that is aware of the context and in which the technology is adaptive, responsive, acceptable, and useful to their needs, habits, gestures and emotions.

To achieve its objective, AmI first tries to identify the characteristics of the context of each inhabitant after recognising the specific tools used by the inhabitant and the activities they perform. With this data, the activity is broken down into several components, levels, and sublevels, which are ultimately recomposed in a coherent manner and in an adequate design that is capable of supporting a certain lifestyle. It is a method reminiscent of that presented in 1936, within the context of architectural standardisation, by the German architect Ernst Neufert in his well-known book *Bauentwurfslehre* (edition in English: Architects’ Data) (Figure 6).

AmI can contribute important technological advances to the human habitat, with particular reference to services for people with specific needs and who may require specific solutions. The advantages of AmI is its adaptability to respond gradually to the level of deterioration that people may suffer due to age or illness (Figure 7). Since its introduction, AmI has become part of the core strategies of many of the world’s leading technology companies.

Based in these concepts, the Ambient Assisted Living Joint Research and Development Programme was launched in the European Union in 2008, as a response to this new technological paradigm. Ambient Assisted Living appeared, for the first time, as a concept related to health, demographic change, well-being, and the design and construction of environments. Later, starting in 2014, this programme was renamed as the Active Assisted Living Research and Development Programme.

Active assisted living, or AAL, applies the tools and methodology of ambient intelligence [25] in order to promote and extend the quality of life of older adults in their environment, promote healthy ageing, reduce social and health system costs of the EU states, and create a framework for the development of common standards with which to strengthen European industry in the area of information technology and communication (ITC) [26]. The three objectives of AAL are:To improve the quality of life and well-being of older adults and their carers through the availability of ITC-based products and services, facilitating their active and healthy ageing;To sustain a critical mass of research, development, and innovation applied to a trans-European scope for ITC products and services, in particular, those affecting small- and medium-sized industries. The intention is to increase private investment and improve the conditions of industrial exploitation;To contribute to the sustainability of health and assistance systems for caring for older adults, establishing a coherent framework for the development of European solutions and foci, including minimum common standards that adapt to the diversity of social preferences, and comply with regulatory aspects.

These three AAL objectives are focused on the elder population; nonetheless, they do not yet distinguish the different characteristics of this collective, among which a huge number of people with cognitive disabilities can be found. The need to reach the next step of developing services and products dedicated to these people is shown, this being a pending issue for development in the near future.

The AAL is being conducted jointly by EU Member States and other countries associated with the Horizon 2020 programme, such as Israel, Switzerland, and Canada. The first AAL-1 programme was effective until 2013 and had six calls for funding. As of 2014, the initiative continued with a new AAL-2 research and development programme (2014–2020) with a budget of around 600 million Euros, managed within the framework of the Horizon 2020 Research and Innovation Funding Programme.

### 4.2. Practical Applications and Research Projects of Active Assisted Living

Over the past fifteen years, the AAL Programme has been focused on addressing the challenges of an ageing population and on taking advantage of the opportunities that this entails. Of the 155 projects financed by the AAL-1 Programme between 2008 and 2013, only 19 projects introduced solutions into the market. This corresponds to an approximate proportion of one project reaching maturity in the market for every 10 research projects financed by the AAL Programme, up until 2021. The examples shown below are those from the review that have had the greatest impact in the scientific literature.

One of the first projects to be financed, which had a notable impact, was the Perspective Spaces Promoting Independent Ageing Project—PERSONA [27]. This project focused on the use of devices through an artificial pet, robot, or object in the decoration of the home, devices with the capacity to interact with the web on behalf of the user, establishing an empathetic relationship with the latter in order to serve them as an intelligent intermediary; however, the most important contribution of the PERSONA Project was the enumeration and classification of the types of services that, in general terms, older adults may require:Services to complement the skills and capacities of older adults in the Instrumental Activities of Daily Living, guiding the user throughout the day, for example, assistance with home chores (cooking, cleaning, washing and shopping), and personal care (hygiene, lifestyle, healthy diet, and reminders to perform activities and take medication);Services to prevent injuries at home, making the occupants feel safe, and giving older adults the capacity to manage their lives in their own space without the need for the constant presence of carers;Social integration services, to alleviate loneliness and isolation, providing a means of communication based on ICT, to satisfy the need for company, exchange experiences, and help to create friendships, social contacts, and opportunities to participate in community activities;Services for mobility, helping older adults to perform activities in their neighbourhood, and encouraging them to go out autonomously with confidence and security, for example, providing information on public transportation, helping with navigation or assisting them when they feel lost.

For an AAL system to be implemented in a given environment, the PERSONA project defined four different levels on which to scale the distribution of services: the person; the home; the neighbourhood or nearby geographical space; and the city or broader geographic space [28]. The first level is aimed at the older adult who, finding themselves in more or less difficult conditions, can directly benefit from a product, service, or system, the characteristics of which provide the user with a direct support function. The second level includes the possibility of adapting the home environment to the specific needs of the user. The third and fourth levels consist of systems, directly or indirectly assisting with social and/or affective matters and which promote inclusivity through services and technologies, which increase the potential for relationships between older adults and their family and friends (Figure 8).

Another project, called the AALIANCE Project [29], involved a more transversal vision based on three spheres of activity:(a)The domestic environment or “ageing at home”: ageing while enjoying a better quality of life for a longer period of time whilst maintaining a higher degree of independence, autonomy, and dignity, with the help of technology. The home environment must be perceived as relaxing, mobility must be viable, and transportation must be available;(b)(The work environment or “active ageing at work”: ageing and staying active by continuing to be useful to society. A person remains productive for a longer period of time with access to ICT via online learning tools, coupled with innovative actions in the workplace, facilitating a better work-life balance;(c)The social environment or “ageing in the community”: ageing while remaining in contact with partners, friends, and family members. A person is socially active and creative in social networks. In addition, he or she has easy access to commercial and public services, reducing social isolation and loneliness, one of the main problems among the ageing population.

Other research projects related to the field of AAL have investigated the problems of urban mobility and accessibility to transportation services to make it easier for older adults to move around the city. The ASK-IT Project [28] focused on the development of applications that could help people with motor disabilities to access public transportation through multimedia and multimodal information services activated according to the geographic location of the user. The project aimed, essentially, to offer transportation guidance and information. Another example of the action of AAL in this sphere is the VADEO [30] application, in which users can indicate problems with accessibility in any space of a city and can be guided through alternate routes.

Using web technology, the AAL Programme has also promoted the creation of gaming networks, entertainment, and leisure options. For example, the SilverGame Project [31] involved developing applications for attractive multimedia games and stimulants to promote a social connection to the provider of a virtual environment through Web 2.0. This environment allows its users to share hobbies, such as singing or dancing, and, ultimately, to share online experiences in order to maintain social contact if they are not able to travel.

Some of the research projects funded through the AAL Programme have resulted in products and services that have ultimately been marketed. Among these, the 2PCS solution aims to offer the user a greater freedom of movement, thus promoting a higher degree of physical security. It is a service that enables the older adult to have more autonomy and to go and walk outside as it keeps them connected with their relatives or carers. Movement sensors, coupled with self-learning algorithms, empower both the end-user and their friends and family in order to provide care by means of quick and anticipated responses thanks to the early detection of risks as a proactive method for the prevention of more serious complications [32]. Another product is SENSARA HomeCare, a platform that enables older adults to live independently and safely for longer. It offers an intelligent senior lifestyle monitoring system with a focus on preventative care and personalised alarm systems [32].

This innovative landscape is made up of a new economic and technological ecosystem in which several actors coexist and interact with one another: inhabitants, users, carers, researchers, technological and integration companies, service providers, social service entities, and public administrations. It is interesting to note how AAL has additionally attracted the interest of architects, urbanists, psychologists, gerontologists, and sociologists. It is thus transforming the initial name of the research programme into a concept, or model, for the design of architectural and urban spaces that the population can use and live in throughout their lifetime. It can be argued that a new residential assistance paradigm called AAL has arisen during the last decade, but this has yet to be theorised or conceptualised.

### 4.3. Development and Future Implementation of Active Assisted Living

The evolution of the notion of assisted living during the next ten years will come about through significant advances in cloud-based service technologies and standard service platforms. The development of the internet of things and e-trade portals will promote new business models that will serve to permeate this stratum of the older population. Services based on a web environment e-cloud will open a new way of incorporating the experience of each user in the cocreation of content. AAL will serve to improve the services of the providers who, in turn, will respond to the specific demands of the users. This should also help public administrations to better plan their services as a means of quickly adapting to changing social needs.

With the correct AAL implementation, all home users will be able to establish personalised solutions, according to their needs, desires, and economic resources as they would with decorating and equipping their homes. However, for this to occur, technological platforms are needed based on the standards that support the interoperability between the different manufacturing and provider services distributed throughout the network. These platforms still have a long way to go in order to reach this technological standard. Four technological requirements in general must also be met [27]:Natural interfaces: interaction systems between people and computers must be intuitive to use, sensitive to the context and multimodal, that is, multiuser, multilingual, multichannel, and multipurpose. They must be based on the most common methods of human interaction, such as voice and gestures. The most paradigmatic example is the Nintendo Wii, which revolutionised the console world, due to the fact that it has a more natural command than other games. As a result, it enables greater immersion, despite its narrative graphics, and despite the fact that its quality is lower than that of the competition;Dynamic networks of massively distributed devices: all devices must communicate through networks, cable or wireless, in such a way that they may process information in a co-ordinated manner. In addition, these networks must be dynamic and able to reconfigure in the presence, absence, or error of a device. An example of this is what happens when one travels from one country to another: this is detected through the network of the destination country. Thus, the user can make calls, or perform other functions, without having to reconfigure their terminal;Comfortable hardware: the integration of devices with processing capacity in everyday objects, such as vehicles, furniture, clothing, toys, home appliances, etc., that are not annoying to use and are transparent. One of the basic lines of work is what is called wearable computing, which is developed in accessories, such as watches, cell phones, and clothing. These devices are equipped with sensors measuring vital signs and sending data or alarms, if necessary;Security and reliability: the entire technological layer of intercommunicated and omnipresent devices must be secured against indiscriminate use. For this, robust systems must have built-in information encryption methods and authentication mechanisms which use, for example, biometry-based techniques (fingerprint, iris, voice recognition, etc.).

The intelligent environment must be capable of maintaining, extending, or adapting to the evolution and changing needs of ageing people as to ensure they have healthier, safer, and more comfortable lives with more personal autonomy and maintain vital contact with their cultural and social environments for as long as possible.

## 5. A Critical Appraisal of the Implementation of Active Assisted Living: Discussion

The great opportunity and the main challenge of the AAL programme has been to establish tools to enable older adults to enjoy the benefits of Health 2.0, in a joint and synergetic manner, with their nearby environments and with their carers [32]; however, the technical options that companies offer today are still far from 100% AAL. This is because the solutions currently being implemented do not yet achieve the benefits and objectives intended to be achieved. Despite promising technological advances, and the fact that current technology is in a position to contribute knowledge to provide some services that improve the quality of life of older adults, AAL remains more of a vision than a reality. AAL is still in its initial stage. So, “Why are its services and products not as available as may be desirable?”, “Why has the model not been developed more broadly?”

The general implementation and development of AAL faces several difficulties [33]. One of the main barriers to the application of AAL is related to how older adults perceive, use, and interpret technology at this time. The capacity of this age group to absorb the intense use of technology distributed and hidden within the environment or in the person’s daily activities is still very limited [34]. In the current technological and digital revolution, society is experiencing a moment of transition, in which several generations, with different levels of technological experience, co-exist: the grandparents, the sons/daughters and the grandchildren. The use of new technologies is expected to advance, in parallel to the carers progressively becoming older as they will have developed digital needs and demands, based on their own life experiences.

Another limiting aspect of AAL is the need for respect for the user’s privacy, which is a significant barrier of an ethical and legal nature [35]. Data privacy is an area of data protection that concerns the proper handling of sensitive data, including, notably, personal information and also other data of a confidential nature. In this field, it is still necessary to improve the legal framework that effectively standardises and protects the use and handling of the information available on the devices that interact with the user in their intimate environment. The balance between surveillance and privacy, the ethical and transparent use of data, which would deserve, in and of itself, a monographic assessment, hinders full confidence in the domestic introduction of technologies that use personal data for the purpose of providing services.

The reliability, safety, and maintenance of technological devices may also cause concern for the user, as, at present, it is difficult to guarantee that all systems will operate consistently and correctly, at all times, and that the failure of the systems will not entail a new ‘technological dependency’. This may be an additional disability to the physical or cognitive dependencies derived from old age, in the sense that, if carrying out certain activities safely depends completely on technology and it fails, the person is prevented from carrying out those activities.

The development of AAL services in professional, social or domestic environments has also been uneven until now. Most of the efforts have focused on creating solutions related to the home environment, such as telemonitoring, measuring, therapeutic compliance, nutritional support, personal safety, and support for daily life activities. The development of services related to professional and social environments, such as the examples described in the previous sections, are, at present, very rare. In general, the implementation of AAL services has been in small-scale pilot projects, with the services almost always focused on home care (to prevent falls, facilitate calls and remote assistance, monitor vital signs, etc.).

This lack of implementation in non-domestic environments has perhaps been due to the fact that specific scales of spatial application have not been contemplated or sufficiently developed, for instance, the medium scale of facilities or the territorial scale of urban planning. Mid-term (four years) and long-term (ten years) forecasts are also needed for homes that are adapted and adaptable and aimed at people who require diverse and personalised needs for each individual sector.

Technical–commercial limitations exist with regard to the processes required for the implementation of devices and the deployment of technology in the domestic environment. Isolated, single-function systems are easy to install, but the complexity greatly increases when several technologies must be combined. In a market controlled by companies with patented products, a market in which corporations aspire to be dominant providers, technological barriers are created by the interoperability of the devices [33].

In contrast, it is worth mentioning the home products available from large companies, such as Securitas Direct or Konica and, most importantly, from the GAFA telecommunication companies. These companies have managed to introduce personal assistants, such as Alexa and Google Home, into the population in a highly effective manner. In the near future, together with televisions and cell phones, these assistants will be the doorway to the services that these firms will be offering to older users. The capillarity that the GAFA has achieved in the market contrasts with the scarce implementation of the products arising from AAL. The EU programme was initiated with the purpose of creating a technological business fabric capable of producing ICT products and services aimed at improving the quality of life of older adults.

However, technological innovation is a necessary yet insufficient condition for a space to meet the vital needs of its users. Developing such spaces involves creating environments and surroundings in which human beings can feel fulfilled, with complete awareness, autonomy, and freedom. A house is not a machine just for living, as Le Corbusier said, but a home to be inhabited that is able to promote personal fulfilment.

People’s quality of life also depends on intangible factors and values that do not depend on technology. The identity, character, and quality of the physical environment are intimately related to the vital, social, and cultural experience of each individual. In other words, the cultural and symbolic components of spaces have a special impact on people’s emotional and cognitive stability. If an older adult can inhabit a personal space and preserve a sense of belonging to a place, with autonomy, they will have a better quality of life. The importance of successful design of space, residences and homes for older adults is to create a sense of belonging to an environment. In each individual’s space where they reside, their history, memories and experiences are interwoven with those spaces where they have lived.

Another, and perhaps the greatest, barrier to complete implementation of AAL is the transformation of the syntagm Active Assisted Living from an international programme into a conceptual paradigm for intervention in domestic and urban spaces, beyond the one-way service-user relationship, by means of a technological product or service. Under the new paradigm, the term extends to the multiple dimensions surrounding people, including cognitive and knowledge aspects, whereby interactions are not only instrumental but also, more importantly, meaningful.

The notion of AAL needs to be updated, but above all, it requires a definition of the environment or surroundings in which people live as well as the implications of the former for the latter. The lack of an articulated theory on what AAL means, as well as its impact on the architecture and planning of cities, is a major limitation that prevents it from being implemented in a sustainable, consistent, and ongoing way.

## 6. Conclusions: Towards Meaningful, Active Assisted Living

Despite the barriers to the application of AAL (technological, institutional, economic, social, and theoretical), architects, urbanists, and engineers have begun to outline a new way of understanding and designing environments to be inhabited, in the mid and long term, by populations which change, evolve, and age. In due course, AAL will be implemented because of the efficiency, comfort, and well-being it entails. The main conclusions obtained in this study are now presented. The conclusions are fully supported by the results obtained as well as those referenced in the secondary literature.

Several conclusions can be drawn from the study carried out on AAL. Firstly, it is shown that directives are necessary in order to establish a legal framework as a means to regulate the implementation of this new technological paradigm. In this way, it would be possible to systematise and order the use and management of information available on home devices that interact with the user within their environment.

Secondly, it should be noted that, in the field of active assisted living, there has not yet been an adequate population segmentation of what constitutes the collective of older adults. The scientific literature consulted and the projects analysed do not currently take into account AAL developments as applied to people with cognitive disabilities. This lack indicates an interesting line of research to be followed and implemented by the competent authorities.

Another conclusion that can be drawn is that almost all the initiatives implemented so far in AAL focus on designing services based on new technologies, with which the technical component and vision are excessively marked. Most of the time, AAL does not address the issue of the nature of physical environments, and domestic and architectural spaces where these emerging technologies have to be integrated. Above all, it seems that the people to whom AAL is directed are forgotten.

As mentioned above in the results, Ambient Assisted Living as a paradigm or design concept must not be limited to providing technology-based services. The reason for this is that, although technology is a catalytic tool for some of the material challenges that society faces, it is insufficient in and by itself to respond to human needs (Figure 9).

The relationship between technological innovation and social or cultural references places architectural design at the centre of the search for solutions, which are a synthesis of needs, behaviours, and interactions between the environment, technology, and people. Architectural design acts as a stimulus for a broad search for solutions that are not just the result of the solving of one technical problem. Architecture, both as an art and a technique, builds the daily scene of the habitat and gives meaning to the places and spaces people inhabit. Through the meaning that architecture gives to spaces and environments, human beings can relate to the values that transcend them as individuals and connect them with their vital, social, and cultural realities [36].

Taking these considerations into account, with the progress of AAL, it will be possible to build assisted living environments that can interact intelligently and emotionally with the inhabitants, enabling them to build their own *exo-brain*. The architectural possibilities are of great interest, in combination with installations, wireless sensor networks, smart grids, the possibilities of retrofitting IoT, improving accessibility, etc. Architecture as a technique will integrate intelligent assistance systems into buildings, embedded within environments in such a way that they may interact invisibly and proactively with the occupants (Figure 10). This will enable the creation of assisted spaces to promote a healthier, safer, and more comfortable life, one in which each person adequately manages and controls their physical, everyday reality with more personal autonomy.

However, architecture goes beyond mere technique or the application of technological systems. Architecture also builds spaces with compositional and formal resources that provide them with a meaning to transcend their intended function. This makes the space comprehensible, where the occupant may find a narrative and symbolic value and a sense of belonging. Architecture reconstructs connections with highly codified and symbolic cultural circuits in such a way that the habitat, in addition to being a refuge and an aid, is, as Roger Bartra says, a cognitive prosthesis of the human being [37].

The challenge that AAL presents as a new paradigm of dwelling consists of knowing how to design spaces that can behave as a kind of collective and individual *exo-brain*, one that is capable of adapting to the gradually changing demands and needs of living associated with the increasing longevity of the population. This challenge also extends to other scales of use, which range from the collective spaces of buildings and facilities to the public spaces of cities.

The main contribution of this work could be specified in that it does not give a definitive answer to the questions raised by that challenge, but provides a series of issues, definitions and data with which to ask new questions from the architectural field. The issues described in this paper improve its value to the discipline of architecture by allowing us to understand AAL as an element to be proactively incorporated into the design of buildings and cities.

In the near future, cities will have the capacity to manage, share, and transfer data and information, in order to facilitate governance decisions, in real time. The conformation of this new city-network model provides a glimpse into the solutions in response to the needs of the older population, which will play a primary and particularly determinant role in city planning. Future intelligent cities will add layers or grids to their configuration, dedicated to sup-porting healthy ageing and providing a better quality of life for people with cognitive dis-abilities. These cities will consist of adapted and adaptable housing as well as facilities and services that can be used by a population that is constantly evolving and demanding new residential solutions. Future urban models, city planning, and housing design must continue to be developed with regard to the inclusion of solutions that integrate this innovative technological character, which is appearing gradually and with increasing speed, into the buildings.

## Figures and Tables

**Figure 1 ijerph-20-05886-f001:**
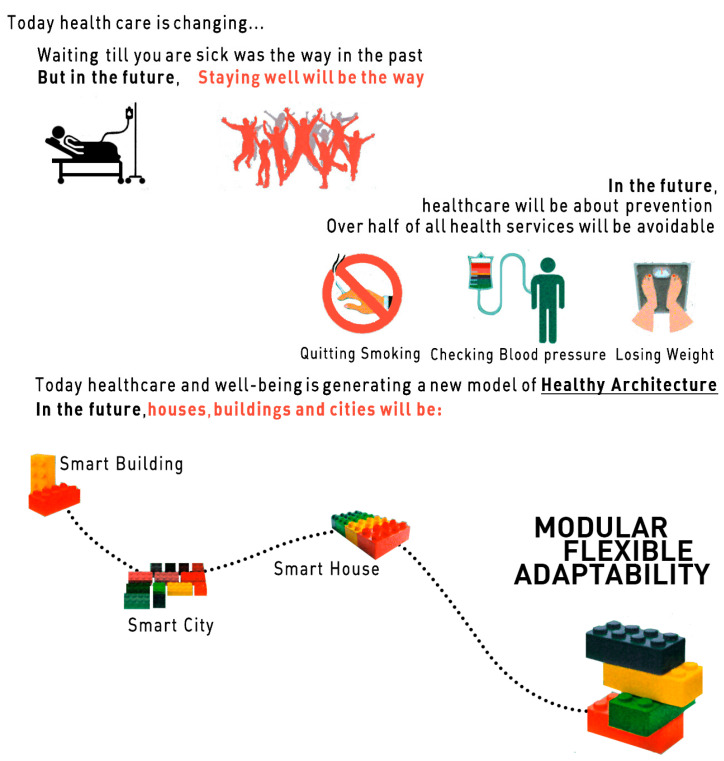
Trends in healthcare and their impact on a healthier architecture. (Source: prepared by the authors and adapted from a graphic by Hyojung Kim and Barcelona Global Design, 2015 [12]).

**Figure 2 ijerph-20-05886-f002:**
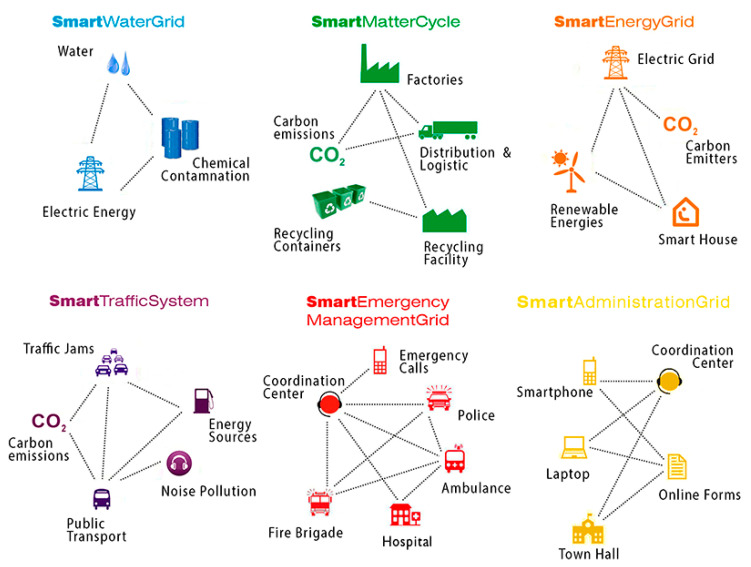
Smart Grids, layers or networks that make up the Smart City. (Source: prepared by the author).

**Figure 3 ijerph-20-05886-f003:**
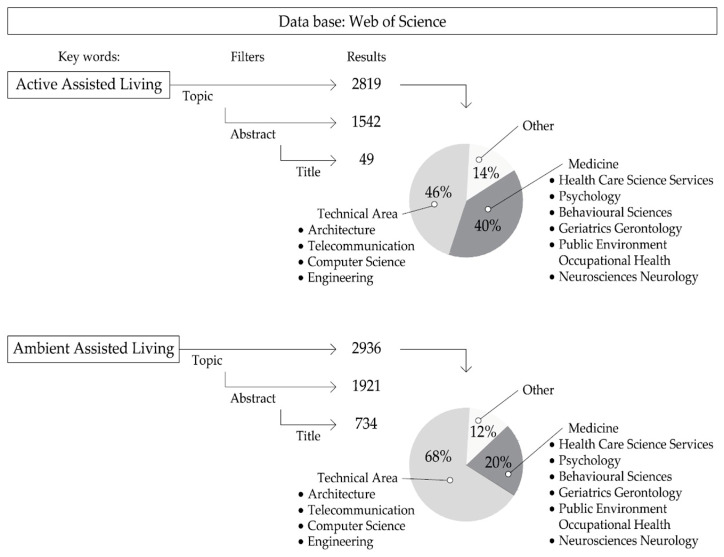
Description of critical literature review methods. (Source: prepared by the authors).

**Figure 4 ijerph-20-05886-f004:**
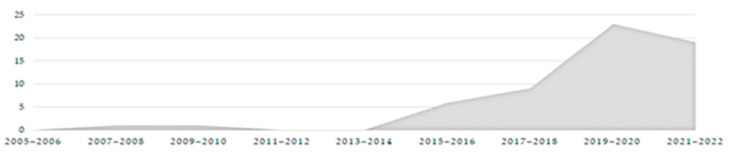
AAL publications results over the years. (Source: Prepared by the authors).

**Figure 5 ijerph-20-05886-f005:**
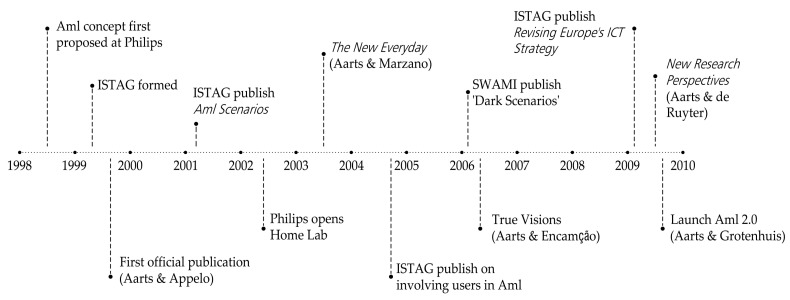
A timeline of key events and publications on Ambient Intelligence. (Source: prepared by the authors and adapted from a graphic by Gunnarsdóttir and Arribas-Ayllon, 2012 [18]).

**Figure 6 ijerph-20-05886-f006:**
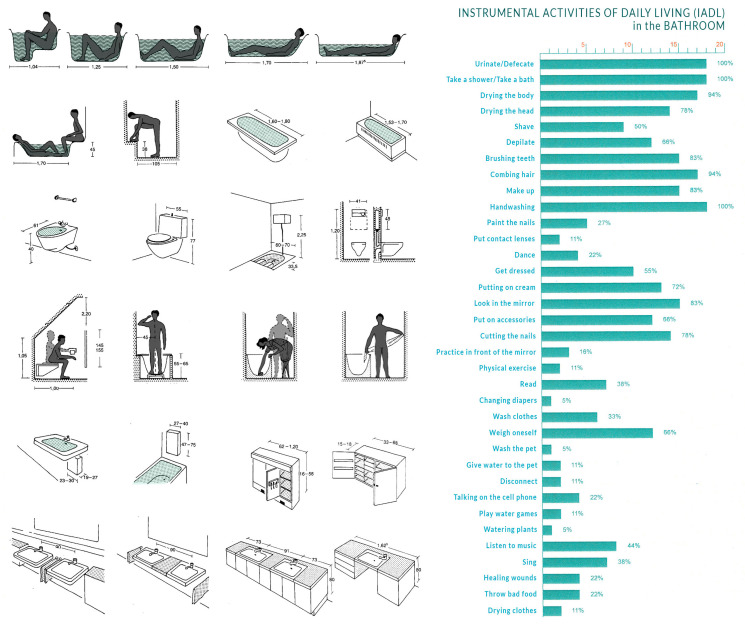
The methods and procedures used by the algorithms of ambient intelligence (AmI) to define the needs of a user are reminiscent of those used by the architect E. Neufert. when he sought architectural standardisation by breaking down human needs into a series of activities. Left: minimum spaces, furniture, and dimensions in the bathroom (Source: Prepared, by the authors and adapted from the book Architects’ Data by Ernst Neufert [24]). Right: Instrumental Activities of Daily Living (IADL) in the bathroom. (Source: prepared by the authors and adapted from a graphic by Martínez Arribas and Roca Sanitarios, 2015 [12]).

**Figure 7 ijerph-20-05886-f007:**
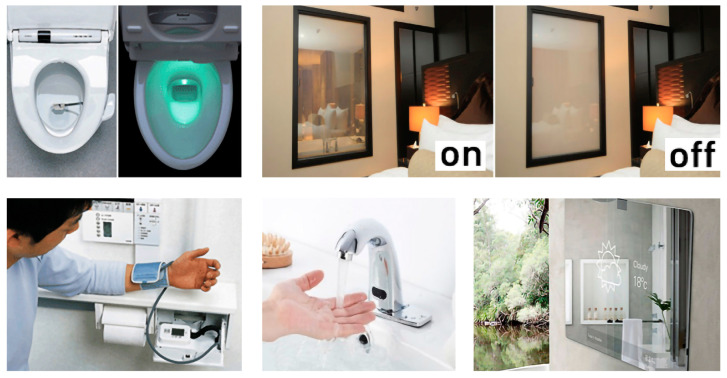
(**Left**): TOTO’s Intelligent Toilet features a urine “sample catcher” that can measure the glucose levels, urine temperature, and hormone levels of women trying to conceive. The wash-let, which also has the standard features of a spray-jet and heated seat, gathers data and communicates with the user’s computer by WIFI, compiling a health report. (**Above right**): Smart mirrors that turn opaque through facial biometric readings are a very useful function in the case of Alzheimer’s patients. (**Below right**): faucets with temperature and water flow sensors and smart mirrors with atmospheric information. (Source: prepared by the authors).

**Figure 8 ijerph-20-05886-f008:**
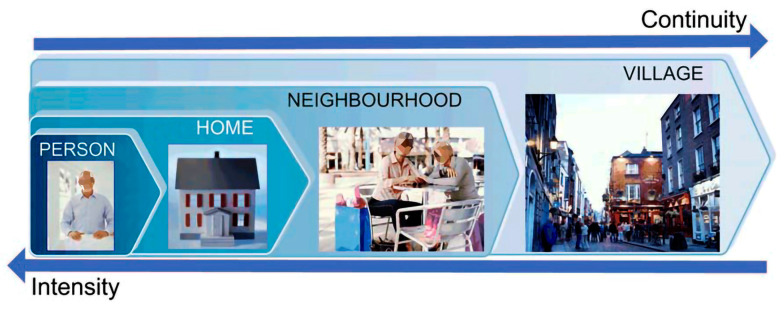
Four levels of AAL services distribution. (Source: prepared by the authors and adapted from a graphic by Guillen and Arredondo, 2011 [28]).

**Figure 9 ijerph-20-05886-f009:**
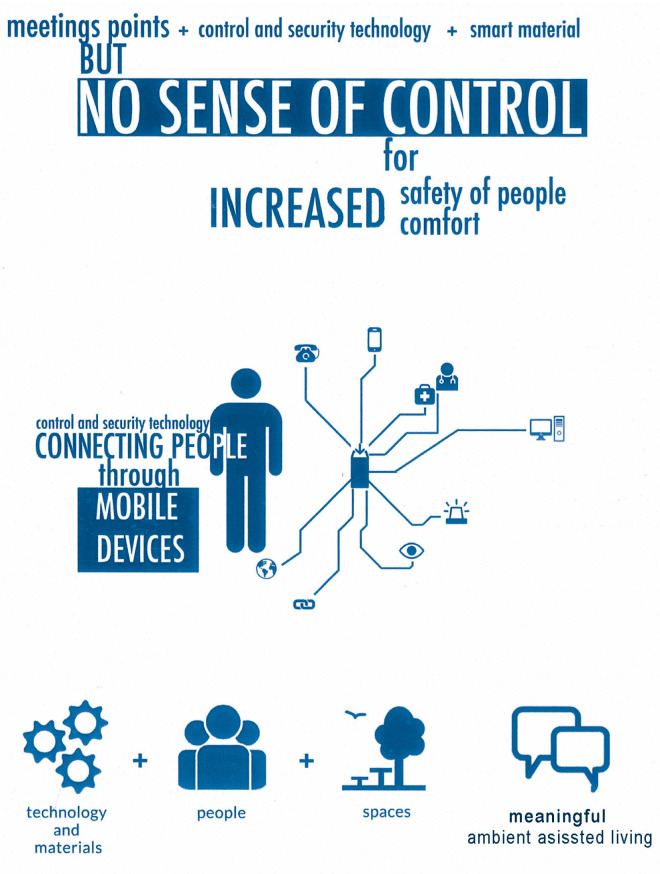
Meaningful Active or Ambient Assisted Living. (Source: prepared by the author and adapted from a graphic by Hernández Perdomo and Iseco Sistemas, 2015 [12]).

**Figure 10 ijerph-20-05886-f010:**
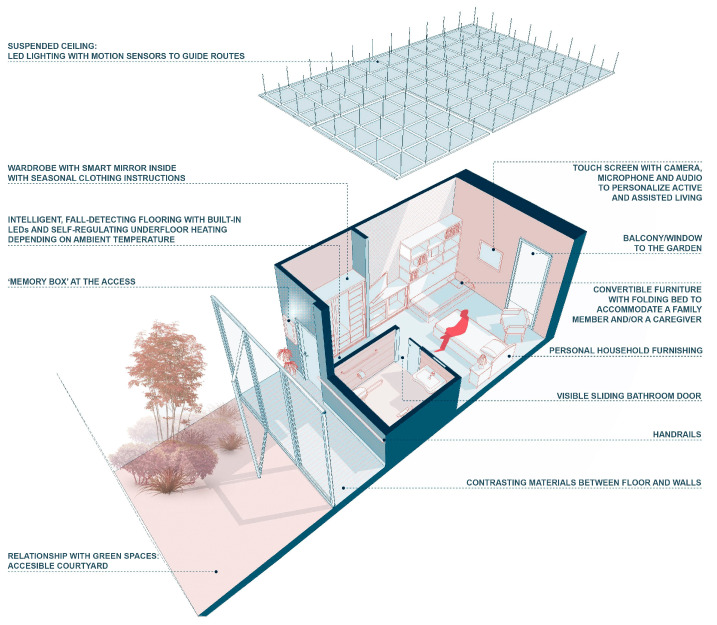
Design of a room for an Alzheimer’s disease patient (Source: prepared by the author and adapted from Thematic Workshop at Politecnico di Milano directed by S. Quesada-García).

## Data Availability

Not applicable.

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
