# Peer review of "Active and Assisted Living, a Practice for the Ageing Population and People with Cognitive Disabilities: An Architectural Perspective"

_ijerph, 2023, doi:10.3390/ijerph20105886_

Round 1

Reviewer 1 Report

Very interesting work. I think the focus of the research is the role that architecture can play in assistive living systems and especially in their acceptance by older people. However, in spite of the title, the research is not focused on research projects related to cognitive disabilities in general (except perhaps for alzhaimer's) so I would suggest a revision of the title or an in-depth look at studies done for other types of cognitive disabilities.

I also point out that the title as presented needs a revision of the English form.

Author Response

Dear reviewer,

Thank you very much for your comments and suggestions; they have been very helpful to us. We have taken them into account while redrafting the manuscript.

It is true that the work is not focused on the group of people with cognitive disabilities, but this is due to, for the moment, AAL programs focusing on the third and fourth age, that is, a group of users who are very generalist and unsegmented. For this reason, there is little scientific literature on AAL dedicated to this group of people with cognitive problems. Somehow, with this work, we want to make this lack apparent, calling attention to a field in which progress must be made, and direct future research efforts. In any case, we have made this specific aspect clearer in the new wording of the text and also in the final conclusions. We have also revised the form of the title; we believe now it is correct.

Thank you again very much for your time and dedication to carrying out this valuable evaluation, which has helped us improve the text.

Reviewer 2 Report

1) In the section "Author Contributions" the original text "For research articles with several authors, a short paragraph specifying their individual contributions must be provided. The following statements should be used" is retained. It can be removed and only the real contributions left.

2) I also think that figures of authors other than the authors of this publication should be redrawn more substantially. However, this is more a matter for the editors.

Strengths:
1. The topic as such.
2. Review nature of the contribution, which is quite valuable.

Weaknesses:
1. Review nature of the contribution only. The paper does not contain "core"
research as such. It is more an appeal to change the attitude in architectural
design towards more senior friendly solutions.
2. The article pretty much "floats on the surface."

Opportunities:
1. The topic of the paper has definitely potential to attract the reader's
attention.

Threats:
1. Extensive use of "cited" graphics, which represents also a kind of threat
in terms of ethics.

Suggestion for major improvement:
In my opinion the authors should narrow their focus to some crucial point and go more in depth. But, what it should exactly be, I do not know - it is a task for authors. However, I do not insist on anything. Perhaps, the images should be re-drawn. 

Author Response

Dear reviewer,

Thank you very much for your comments and suggestions; they have been very helpful to us. We have taken them into account while redrafting the manuscript.

We have removed the final sentence in Author Contributions, which we had missed; thanks for pointing it out. On the other hand, in the manuscript, we have redrawn and substantially modified Figures 1 and 9, which belong to other authors who are duly cited as sources in the photo captions. The left part of Figure 6 comes from the classic book by E. Neufert, and we believe that this illustration is better in its original form because it is from the year 1936. With this type of drawing, it clearly shows the antiquity of working similar to how AI or AmI algorithms are working now.

Thank you again very much for your time and dedication to carrying out this valuable evaluation, which has helped us improve the text.

Reviewer 3 Report

Thank you very much for the interesting contribution. The topic is timely and highly relevant.

The main difficulty with the article is the structure, which deviates from the classic structure (introduction - methods - results - discussion - conclusion). In principle, this is an option, but in this case it unfortunately leads to ambiguities.

The methodology is appropriately chosen, but not stringently carried out. Even with a qualitative approach, a structured analysis is absolutely necessary. The main findings should be described in such a way that they are quickly and clearly recognisable to the reader and it must be comprehensible from which set of analyses the findings result.

I will now list a few points in order that should be revised:

-        The abstract does not list the key findings of the study.

-        The introduction is clearly written. However, it is not presented precisely enough where the concept of AAL originally comes from. (line 57-60)

-        In lines 76-86, the structure is explained. Here it becomes apparent that the connections between the individual paragraphs are not clear. It is undefined whether the "main idea" described in the fifth paragraph is a result of the literature study.

-        Lines 94 - 96 are difficult to understand. Please rephrase.

-        The concepts and keywords (line 106) that were searched for in the databases are not mentioned in the text.

-        In addition, a description of the articles that were focused on in more detail is missing. Are there important standard works that were evaluated? How were the articles selected if the keyword AAL did not lead to satisfactory results?

-        From line 124 onwards, results are described that do not belong to the description of the methods, but should be described as a result of the literature study in a separate chapter.

-        Figure 2 and Figure 3 should be mentioned first in the text before they are shown.

-        The whole of paragraph 3 seems like part of the introduction. Here it becomes unclear whether these are results of the literature study or a general introduction to the topic. Statements are made that are not related to the literature study and are not sufficiently supported by sources.

-        Chapters 4, 5 and 6 are also not clearly related to methodological approaches. Are they results of the literature study or a compilation of the state of knowledge? My impression is that it is a mixture of introduction and discussion.

-        Chapters 8 and 9 present results that are very interesting in terms of content, but whose derivation is unclear. Therefore, they cannot be assessed as scientifically well founded.  

In summary, this is a study that discusses important theories and developments for architecture, but unfortunately has clear weaknesses in the scientific elaboration. I would urgently recommend that the structure of the article be fundamentally revised and that the correlations between the individual chapters be made clear.

Author Response

Dear reviewer,

Thank you very much for your comments and suggestions; they have been very helpful to us. We have taken them into account while redrafting the manuscript. Thank you also for your valuable constructive criticism that allowed us to substantially improve the initial text.

In this sense, we have restructured the entire work to adapt it as much as possible to a more classical structure so the reading of the text does not lead to ambiguities. We hope that this new version will be clearer and more specific, and that, as a whole, it will be better understood.

Now we have included the main result we achieved in the abstract, which is the development that AAL will have in the next 10 years, as well as its influence on the design and construction of buildings in the near future.

We have specified the origin of the AAL concept (lines 66 and 67) in the introduction, and we have also modified the final paragraph of this introductory chapter, where the structure of the article is explained, so that the main idea of the work is better understood (lines 83–98). We believe that the idea is now more clearly transmitted.

The old lines 94–96, which are now lines 269–271, have been redrafted. We hope that now they can be better understood. The keywords searched in the databases have also been included (lines 283–286). The selection of the articles are now listed by those with a greater scientific impact collected in the main databases (lines 277–278).

We have moved the part that previously mentioned the results in Methods to the corresponding chapter, and we have also improved the wording to make it clearer.

On the other hand, we have moved Figures 2 and 3 (now they are Figures 4 and 1, respectively) so that they appear after being mentioned in the text.

Both Chapters 3 and 4 have been moved, because they are part of the background and context in which the work is framed. With this new structure, we believe that the general exposition is clearer and better understood. Likewise, we have restructured Chapters 5 and 6, including them under the same chapter with results. The current Chapter 4 also incorporates a third and new epigraph as a climax to the results of this critical review. This last section shows the development of AAL in the coming years.

The last chapter, now Chapter 6, is clearer and presents the conclusions in a more concise and coherent way, which are supported by the results previously reached.

We hope that with this thorough revision and all the changes made that the manuscript will be better, and that, in this way, the different parts will be better correlated and the content better transmitted.

Thank you again very much for your time and dedication to carrying out this valuable evaluation, which has helped us improve the text.

Reviewer 4 Report

This paper investigates a very common question in the field of architecture: how the architectural practice should respond to technological innovations. Obviously, technology and systems have positive effects on the growing population of the elderly. However, in an architectural context, this should be discussed with both physical and virtual space that covers user experience. There are several sources in the literature about this discussion that should be added to this paper. Please see my further comments below:

1. Introduction is repetitive and fails to present the rationale of this research study. It needs to be elaborated by further sources which can underpin the methodology. 

2. Research question and the aim of this study are not clear, although different aims are mentioned several times within the paper.

3. Language is clear and understandable.

4. This work can benefit from further clarification: with a limited reference list, it explains several features of the topic but cannot direct them to architectural discourses.

5. The method section mentioned a systematic literature review but the results are limited to some percentages. Which topics emerge after literate review? What were the criteria to decide how these sources from several disciplines contribute to the architecture discipline? so on.

6. Line 416-417  "The examples shown below are those that, in the review that has been carried out, have had the greatest impact in the scientific literature." Comments like this do need a reference, how these examples have a great impact, and what were the others. how do you define great impact, etc? They raise more questions than a result. Maybe presenting these examples as a table with their findings, references, and contributions can clarify this chapter.

7. There are huge leaps between scales of product design and urban design throughout the manuscript which cannot reveal how are they related to the architecture and interiors.

8. Service design is essential in this topic but it overshadows architecture. The paper is too long and cannot reveal a cleat rationale, methodology, and results.

Author Response

Dear reviewer,

Thank you very much for your comments and suggestions; they have been very helpful to us. We have taken them into account while redrafting the manuscript. Thank you also for your valuable constructive criticism that allowed us to substantially improve the initial text.

In this sense, we have restructured the entire work to adapt it as much as possible to a more classical structure: introduction, background/context, methods, results, discussion, and conclusions—a structure whose reading does not lead to ambiguities. We hope that now the new text has a clearer structure and is better understood.

We have reorganized the introduction to avoid being repetitive, also modifying the final paragraph of this chapter where the structure and objective of the study are explained (lines 83–98). We hope it is now clearer.

One of the results we reached in this work is that in the architectural field, there are very few references and bibliographies that mention the AAL concept despite the impact this technology has on house and building design. There are still no canonical books or publications that reflect on AAL from an architectural point of view. One of the intentions of this work is to build an epistemology that, up to now, does not exist in the architectural field in relation to AAL.

The review of this paper is a critical review, not a systematic review. In this sense, we seek to identify the most significant items in the field and present a conceptual contribution that can derive new theories. On the other hand, a systematic review aims exhaustive searching to draw together all known knowledge on a topic area; in this case, the review has to be limited due to the lack of publications in the field rather than presenting the conceptual innovation a critical review can provide.

The explanation for the selection of examples has to do with their impact on the scientific literature: those publications that have a high number of citations and are also included in the WoS, Scopus, PubMed, and Science Direct (lines 276–277). In addition, we have included projects with impact as those that have managed to introduce solutions to the market (lines 439–445). We explained that, of the 155 projects financed by the AAL programme, only 19 projects have brought products and services to the market, and of these, very few have generated scientific literature on the matter. Among the few that have done so, we focused on those who have proposed an epistemology that serves as the basis for the construction of a theory in this regard. The intention of this article is to serve as a starting point and theoretical basis for future AAL research in the architectural field.

One of the results of this article is that it remains to be seen how AAL is going to influence architecture and building interiors, and precisely one of its most negative aspects is that the new paradigm of AAL focuses on the technological component and services, leaving aside aspects of architectural design and construction. This factor is one of the points that we want to highlight in this work, which is that the development of AAL is not possible without considering its integration into architecture and without addressing the meaning that the architectural discipline gives to environments and spaces.

We hope that with this thorough revision and all the changes made that the manuscript will be better, and that, in this way, the different parts will be better correlated and the content better transmitted.

Thank you again very much for your time and dedication to carrying out this valuable evaluation, which has helped us improve the text.

Round 2

Reviewer 3 Report

The revised article has improved significantly compared to the first version. The new structure is clearer and easier to understand.
I have no further comments.

Reviewer 4 Report

Your revisions and responses are appreciated.